# Foreign Trade and Income Convergence in Latin America

**Evânio M. Paulo** [1,*,†] and **Osmar T. Souza** [2,†]

1  Department of Economics, Federal University of Roraima, Ave. Ene Garcez, 2413, Boa Vista 69310-000, Brazil

2  Business School, Pontifical Catholic University of Rio Grande do Sul, Porto Alegre 90619-900, Brazil; osmar.souza@pucrs.br

\*  Correspondence: evanio.paulo@ufrr.br; Tel.: +55-95-98113-0578

†  These authors contributed equally to this work.

**Abstract:** This paper explores the variation in inequality as a measure of convergence or divergence in per capita income. It is proposed to decompose this variation into different selected categories of foreign trade, especially due to technological intensity and intraregional and interregional trade. The first category of selection helps to understand how the different stages of development of foreign trade are associated with the convergence of per capita income between nations. The second perception shows whether regional integration actions have had any effect on the region's convergence. As a result, there is evidence of absolute convergence for the sample of Latin American countries and territories selected in the 1995–2017 period. The convergence was on the order of 16.7% and became more intense after 2004; however, it decelerated in the most recent period which is characterized by a phase of lower growth. The important participation of foreign trade in determining regional asymmetries in Latin America is confirmed and a significant effect of intraregional trade in the sense of reducing inequality.

**Keywords:** Latin America; foreign trade; decomposition

## 1. Introduction

Given the considerable influence foreign trade has on the pattern of regional development in Latin America, many characteristics of the asymmetries among the countries in the region are closely linked to the structure of foreign trade. Hence, in the same pattern of trade that forged part of the development in Latin American, there are also elements that explain the distortions within the region's productive structure.

Furthermore, international trade relations have been marked by significant changes, including China's growing importance as a trading power. Latin America, as noted in the literature, has also felt the impacts of China's rise, such as in the rapid increase in commodity prices in international markets. Another important effect is the increased competition between local and Chinese manufacturing industries, which has caused changes that, as of yet, are not entirely clear Cunha et al. (2011).

These changes in global trade patterns have impacted Latin America in different ways. In the specific case of China's role, Cunha et al. (2011) highlighted the existence of at least two trade patterns: that identified with the Mexican and Central American experience, where trade deficits predominate; and that seen in the countries of South America, where there have been periods of surpluses sustained by the boom in exports of primary and energy commodities, as well as in low-tech manufactured goods.

When attempting to analyze these effects, two elements are implicit and central to this objective. The first consists of determining the pattern of income convergence or divergence between countries, the dimension of which gives an idea of the degree of regional dispersion of national incomes in relation to the inhabitants (product per capita). As is demonstrated in the appropriate sections below, herein, an alternative approach to the traditional analysis of convergence is proposed, one in which, in addition to the dynamics

of convergence or divergence, the degree of structural inequality between the economies is considered.

The second element is, of course, to observe what effects foreign trade and its peculiarities have on this dynamic. Once again, a set of elements is proposed that characterize different perceptions of foreign trade in Latin American countries. Given the possibilities offered by the proposed alternative analytical approach to income convergence, it is possible to decompose, according to these different perceptions, the variation in structural inequality or, in other words, the variation in the dispersion of national income ($\sigma$-convergence) over time.

For this purpose, this paper is divided as follows. Following this introduction, in Section 2, the techniques applied in the analysis of convergence or distribution of national income and its decomposition are systematized, and the key concepts required for an understanding of the paper are presented. Section 3 provides an overview of foreign trade statistics in Latin America, which provides an initial characterization of foreign trade relations, where the elements necessary to understand the analysis of the results. And, finally, the results and discussions are presented in Section 4.

## 2. Convergence and Decomposing Inequality

The analytical model of income convergence consolidated in the literature involves regressing income growth rates to test whether poor countries grow faster than richer countries. However, this approach provides little information on the degree of income dispersion between countries. Additionally, according to O'Neill and van Kerm (2008), one might observe poor countries with higher growth rates than rich countries and, even so, the incomes diverge.

Moreover, traditional analytical models of convergence are restricted with respect to the possibility of decomposing the distributional variations to infer which variables induce changes in the convergence dynamics of the product per capita between countries or regions. These restrictions ultimately impair the understanding of how structural changes and economic policy cycles affect inequality and, consequently, convergence.

Based on this model of analysis of convergence through the regression of the growth rate, the two common parameters most widely used in the literature to infer information about the reduction in income disparities between countries are developed. The first refers to a measure that expresses to what extent poor economies tend to grow faster than rich ones ($\beta$-convergence). The second expresses whether the dispersion in the real levels of product per capita tend to decrease over time ($\sigma$-convergence).

The analytical model proposed in this paper follows the specifications of O'Neill and van Kerm (2008), in which the traditional dispersion measures for ascertaining the degree of asymmetry of the product per capita among a group of countries, such as the standard deviation, are replaced by the Gini index. This modification ensures greater versatility when analyzing convergence, allowing greater inference and analytical strategies associated with the Gini index.

Thus, the approach proposed here enables important inferences about the degree of divergence, unlike traditional measures that present results, mainly associated with its variation. Another advantage of adopting an approach based on the concentration index is the possibility of decomposing variations by sources, which allows greater flexibility in the analysis of convergence and the inclusion of new parameters.

The authors associate the variation of the generalized Gini coefficient between the initial period and the final period, $\Delta(v)$, as a measure of "$\sigma$-convergence". Additionally, this variation can be decomposed into factors, namely, $R(v)$, which measures the degree of reclassification of the observations making up the sample, and $P(v)$, which is a measure of the progressive income growth. The latter measure can be interpreted as a parameter associated with "$\beta$-convergence".



### 2.1. Decomposing the Generalized Gini Coefficient

Inequality is measured using the generalized Gini coefficient, G($\chi$; $v$), where $\chi$ expresses the distribution of income in a given period and $v$ is a parameter associated with sensitivity or aversion to inequality, placing greater weight on the differences between the poorest observations. The traditional Gini coefficient is a particular case of the generalized coefficient when $v = 2$. The decomposition of the generalized coefficient takes the following form:

$$\Delta(v) = R(v) - P(v) \tag{1}$$

where $\Delta(v) = G(\chi^1; v) - G(\chi^0; v)$, that is, the variation of the generalized Gini coefficient between the initial period and the final period. $R(v)$ is a measure associated with the reclassification of observations in the data set, that is, this parameter can be interpreted as a measure of mobility, according to Yitzhaki and Wodon (2009). Finally, $P(v)$ can be interpreted as an indicator of how much growth proportionately benefited the observations at the lower end of the distribution in the initial period, according to Jenkins and Van Kerm (2009). Thus, these parameters can be expressed by the following equations:

$$R(v) = G(\chi^1; v) - C(\chi^0, \chi^1; v) \tag{2}$$

$$P(v) = G(\chi^1; v) - C(\chi^0, \chi^1; v) \tag{3}$$

where $C(\chi^0, \chi^1; v)$ is the generalized concentration coefficient from period 1 to period 0. Applying this notion to the convergence analysis, the variation in the generalized coefficient, $\Delta(v)$, is associated, as mentioned above, with a measure of variation in the distribution of product per capita among a set of countries. This notion is precisely the concept of $\sigma$-convergence. The notion of $\beta$-convergence, on the other hand, expresses whether the growth rate of the product per capita of the poorest countries tends to grow faster than that of the richest countries, which corresponds to the effect captured by $P(v)$.

There is still some debate in the literature regarding which of these parameters best expresses the convergence or divergence of income, as well as about how they should be calculated and analyzed. For example, in Lerman and Yitzhaki (1985), there are arguments suggesting the analysis of convergence should focus on diminishing the variance between countries ($\sigma$-convergence). An alternative view, however, is found in Sala-i-Martin (2006), who states that convergence analysis should be based on the two dispersion parameters. Thus, both concepts of convergence are associated and must be empirically analyzed together.

Regarding the calculation of the Gini coefficient applied to the notion of convergence analysis, it measures twice the area between the 45-degree line, or perfectly asymmetric curve, in which each country has exactly the same product per capita, and the Lorenz curve of the real asymmetry of the regional productive structure. Mathematically, it can be expressed as follows:

$$G = 2 \int_0^1 [\rho - L(\rho)] d\rho \tag{4}$$

where $\rho$ is the proportion of countries in relation to the total number of countries in Latin America, and L($\rho$) denotes the proportion of the total Latin American product per capita corresponding to each country. The above equation can also be expressed in terms of the cumulative distribution function, $F(x)$, and the yield density function, $f(x)$, as follows:

$$G = 1 - 2 \int_a^b [1 - F(x)] \frac{x}{\mu} f(x) dx \tag{5}$$

where $a$ and $b$ are, respectively, the highest and lowest product per capita and $\mu$ is the average product per capita. From the second formulation, it is clear that the Gini coefficient is equal to one minus two times the weighted average of the normalized average yields (Lambert 1992). The weights given by $[1 - F(x)]$ are determined by the relative classification

of the agents' income in the distribution. The lowest income receives a weight of one; weights decrease as one moves towards the highest income, which receives a weight of zero (Jenkins and Van Kerm 2009; O'Neill and van Kerm 2008).

Under conditions of bivariate distribution of income in two periods, period 1 and period 2, an analogous concept can be defined. Aligning the countries in ascending order of product per capita in the first period, the concentration curve plot $\rho$, the distribution of countries, against $C(\rho)$, the proportion of product per capita in the second period, as defined below:

$$C_1^2 = 2 \int_0^1 [p - C(x)]dp \tag{6}$$

where $G_1^2$ is a concentration coefficient of product per capita, representing the difference between the perfect equality line (the 45-degree line in the Lorenz's diagram) and the real distribution function of per capita product. By integrating the above equation, the concentration coefficient, $G_1^2$, is determined by a weighted average of the normalized average income for period 2, where the weights are determined by the relative classification of period 1, as follows:

$$C_1^2 = 1 - 2 \int_a^b \int_a^b [1 - F^1(x)] \frac{y}{\mu^2} h(x,y)dxdy \tag{7}$$

Here, $F^1(x)$ is a function of cumulative distribution of the income for period 1, $\mu^2$ is the square of the average normalized income for period 2, and h($x,y$) is the function of bivariate density of income in periods 1 and 2. At this point, considering the above equations, the decomposition structure of the Gini index proposed by Jenkins and Van Kerm (2009) and O'Neill and van Kerm (2008) is used, which shows that the variation of the index, $\Delta G$, in periods 1 and 2, can be decomposed into two components, as follows:

$$\Delta G = G_2 - G_1 = (G^2 - C_1^2) - (G^1 - C_1^2) = R - BC \tag{8}$$

The variation in the Gini index, $\Delta G$, measures the evolution of the degree of inequality between two different periods. Thus, when applied to the context of countries and regions, it expresses a direct measure of $\sigma$-convergence. In the second term, $BC$ is the weighted average of product growth per capita in each country, in which weights are given according to each country's classification in the initial income distribution, as per Jenkins and Van Kerm (2009). A $BC$ greater than zero implies a reduction in inequality, in addition to meaning that growth rates are relatively high among the poorest countries. This can be interpreted as a measure of $\beta$-convergence.

Additionally, when an initially poorer region succeeds in surpassing one that was richer, an increase in the growth rate of the region, which was previously poorer; now increases the inequality. Thus, when observing this effect, Friedman (1992) noted that $\beta$-convergence can thus be counterbalanced. $R$, which can be defined as the residual difference between $\Delta G$ and $BC$, quantifies the effect of offsetting the reclassification on the reduction in inequality (a measure of mobility between countries) (Jenkins and Van Kerm 2009).

Decomposing the Generalized Gini Coefficient: An Extension

An understanding of the composition of $\sigma$-convergence may provide an indication of the disparities among the Latin American economies. Therefore, this section seeks to formulate an operational concept that can be structured as an algorithm to decompose it by its sources. This begins with the equation of the basic identity of demand that should provide a good parameter for this purpose. Thus, our analysis starts from the following equation:

$$Y_{it} = C_t + I_t + G_t + X_t - M_t \tag{9}$$

where $Y_t$ corresponds to the total aggregate product of Latin America in the period $t$; $C_t$ represents the aggregate consumption; $I_t$ the total capital formation; $G_t$ government sector spending; $X_t$ are aggregate exports; and, finally, $M_t$ expresses total imports. A problem arises immediately: the simple composition and comparison of the aggregate demand of the different economies in Latin America is insufficient to discriminate the structural differences between the countries in the region, since the absolute magnitude of each economy says nothing about productivity, which would require a relative approach.

A more efficient model would be its version in terms of units per capita. This version of aggregate demand normalizes the effects of economies' magnitudes and evaluates aggregate demand in relation to population size. Consequently, when Equation (9) is divided by the population size in each period, it takes on the following format:

$$y_t = c_t + i_t + g_t + x_t - m_t \tag{10}$$

where each element of Equation (10) corresponds to its counterpart in Equation (9), only now in terms of units per capita. Thus, the distribution of the formation of aggregate demand provides an operational and relative parameter with which one could approach the measurement of the structural differentials among the Latin American economies. Considering $y_t$ as per capita income, we can calculate the Gini index as a measure of $\sigma$-convergence and decompose it by component of aggregate demand O'Neill and van Kerm (2008).

In this approach, the variables $x_t$ and $m_t$ play a crucial role, as they indicate how foreign trade variables relate to $\sigma$-convergence. To make the analysis more sophisticated, these two variables are disaggregated in two ways: (i) Foreign trade, $x_t$ and $m_t$, by technological intensity, as the theory of dependency portrays Latin America as historically exporting primary products and importing manufactured goods; through this, the aim is to investigate how this pattern affects income convergence or divergence in the region; and (ii) Foreign trade, $x_t$ and $m_t$, by destination markets, as the region has seen attempts at regional integration that seek to enhance intraregional trade as a strategy to overcome the historical trade dependencies characterizing the region; through this lens, the investigation endeavors to ascertain whether this form of intraregional trade can indeed exert an impact on income convergence or divergence within the region.

Since total demand is the sum of aggregate demand in each country, a concentration algorithm can be obtained based on specific methodologies of income distribution Hoffmann (2009). In this case, the total income would correspond to the total aggregate demand. Thus, considering that $y_i t$ is the aggregate demand of the $i$-th country in a total formed by $n$ countries and that the aggregate demands are ordered in such a way that $y_1 t \leq y_2 t \leq \ldots \leq y_n t$. Taking the average given by Equation (11) below, and aggregating the countries from the poorest to the $i$-th position in the series, the cumulative proportion of countries is $p_i = i/n$; the respective accumulated proportion of aggregate demand will be given by Equation (12):

$$\mu = \frac{1}{n} \sum_{n=1}^{n} X_i^2 \tag{11}$$

$$\Phi = \frac{1}{n\mu} \sum_{j=1}^{i} X_i^2 \tag{12}$$

To demonstrate the formulation of the decomposition of the variation, consider Equation (13), where $P_i$ expresses the cumulative proportion of countries and $\Phi_i$ represents the cumulative proportion of income. Additionally, since the total income is a sum of the various sources, it is possible to decompose the contribution of each component to the variation in the Gini or $\sigma$-convergence:

$$G = \frac{1}{n} \sum_{n=1}^{n-1} (P_i - \Phi_i) \tag{13}$$

Since the ordering of income as defined above is maintained, using $y_{hi}$, which expresses the source $k$ of country $i$, the concentration curve of the $k$-th portion can be obtained, which shows how the accumulated proportion of this portion varies depending on of the overall cumulative proportion. Additionally, $\beta_i$ can be represented by the area between this curve and the abscissa axis. Thus, the concentration curve for source $k$ can be expressed as in Equation (14). As $\Phi_i$ is the share of the $k$-th portion in the total aggregate demand, it is possible to express the Gini, as defined in Equation (13), in terms of the sum of the accumulated proportion of income and the concentration coefficient of the $k$ portion, as follows in Equation (15):

$$C_k = 1 - 2\beta_k \tag{14}$$

$$G = \sum_{i=1}^{n} \Phi_i C_k \tag{15}$$

In the context of the decomposition of the variation, the contribution of the $k$-th portion to the change in the total index can be expressed through the associated parameters, namely, coefficient of concentration of the source, $C_k$, and share of the source in the income, $\Phi_i$. Where the first term of the sum on the right side in Equation (16) represents the composition effect and is associated with the change in the participation of a given source. In turn, the second term of the sum on the right side in Equation (16) expresses the concentration effect and represents the change in the total concentration coefficient that results from a change in the particular concentration of the source. Finally, the total effect is expressed with the sum of the composition effect and concentration effect associated with each source, as follows in Equation (17):

$$\Delta G_k = (\bar{C}_k - \bar{G})\Delta\varphi_k + \bar{\varphi}_k \Delta C_k \tag{16}$$

$$\Delta G = \sum_{k=1}^{5} (\bar{C}_h - \bar{G})\Delta\varphi_k + \sum_{k=1}^{5} \bar{\varphi}_k \Delta C_k \tag{17}$$

Thus, this methodology would have the advantage of revealing how a percentage change in consumption levels, capital formation, public sector spending, and net exports affects the structural inequality between Latin American countries, which thus provides a perception of how the distribution of aggregate demand by source and between countries on the continent impacts structural differences between them. As this paper holds a distinct interest in variables related to foreign trade, the utilization of demand equation components as sources for convergence decomposition is warranted. Furthermore, the segmentation of foreign trade by technological intensity and destination of goods finds justification in the theoretical insights concerning the effects of Latin American foreign trade on its developmental trajectory.

### 2.2. Foreign Trade Categories and Source Data

Yitzhaki and Wodon (2009) and Portes (2009) presented the decomposition schemes for the variation in inequality by components. One can also observe that the pattern of foreign trade has a significant effect on the indicators of structural inequality between Latin American nations. The objective now is to open foreign trade accounts to observe how the various subcategories of exports and imports impact this inequality, now analyzed from the perspective of income convergence.

Thus, it is opportune to define which criteria can be used to open regional foreign trade accounts. Based on this perspective, two phenomena stand out in the historical context of Latin American trade relations. The first refers to the predominance of raw material exports, which have historically characterized Latin American trade relations. The second refers to attempts at integrating intracontinental trade. Hence, these two components guide the criteria for opening foreign trade accounts in this paper. In Appendix A Table A3, the categories of exports and imports that reflect these criteria are summarized.

The data were compiled so as to separate the contribution of each selected category towards the total exports of goods. For this reason, it is possible to measure the effect of each of these categories using the decomposition analysis proposed by Yitzhaki and Wodon (2009) and Portes (2009). Thus, it will become apparent how the pattern of foreign trade, based on low-tech products, and trade integration arrangements in regional subspaces impact regional convergence.

With this, we seek to capture how the evolution of intraregional trade, which is associated with the results of regional integration projects between countries, has contributed to the convergence or divergence of the continental product per capita. As trade flows become more intense, intraregionally, there is, as a counterpart, a tendency to accentuate divergences in national patterns of product per capita.

Another feature of special interest in this paper refers to the pattern of raw material exports in Latin America. Thus, Latin American exports and imports are grouped into categories according to technological intensity (see Appendix A Table A3). Thus, the aim is to assess the contribution of the primary export pattern that characterizes some regional. For more information on the classification of exports and expressions by technological intensity, see Table 1 and Lall (2000).

**Table 1.** Technological Classification of Exports.

| Classification | Examples |
|---|---|
| Primary products | Fresh fruit, meat, rice, cocoa, tea, coffee, wood, coal, crude petroleum, gas |
| Manufactured products | |
| Resource-based manufacturing | |
| Agro-/forest-based products | Prepared meats/fruits, beverages, wood products, vegetable oils |
| Other resource-based products | Ore concentrates, petroleum/rubber products, cement, cut gems, glass |
| Low-technology manufacturing | |
| Textile/fashion cluster | Textile fabrics, clothing, headgear, footwear, leather manufactures, travel goods |
| Other low technology | Pottery, simple metal parts/structures, furniture, jewellery, toys, plastic products |
| Medium-technology manufacturing | |
| Automotive products | Passenger vehicles and parts, commercial vehicles, motorcycles and parts |
| Medium technology process industries | Passenger vehicles and parts, commercial vehicles, motorcycles and parts Synthetic fibres, chemicals and paints, fertilisers, plastics, iron, pipes/tubes |
| Medium-technology engineering industries | Engines, motors, industrial machinery, pumps, switchgear, ships, watches |
| High-technology manufacturing | |
| Electronics and electrical products | Office/data processing/telecommunications equip, TVs, transistors, turbines, power generating equipment |
| Other high-technology | Pharmaceuticals, aerospace, optical/measuring instruments, cameras |

Source: Lall (2000).

Finally, regarding the period of analysis and the selected countries, the paper includes a sample of 39 Latin American nations and territories, as illustrated in Table 2, for the period comprising the years 1995 to 2017, using the most recent data from the United Nations Conference on Trade and Development (UNCTAD) databases UNCTAD (2023). This organ is linked to the United Nations (UN) and compiles national foreign trade statistics and estimates. It provides foreign trade measures, national accounts, and other indicators for a

large sample of countries and has been widely used in studies, especially those involving foreign trade.

**Table 2.** Group of selected Latin American countries.

| Latin America | | | |
|---|---|---|---|
| Aruba | Barbados | Granada | Montserrat |
| Anguilla | Chile | Guatemala | Nicaragua |
| Argentina | Colombia | Honduras | Panama |
| Antigua | Costa Rica | Haiti | Peru |
| Bahamas | Cuba | Cayman Islands | Jamaica |
| Paraguay | Belize | Dominica | Saint Kitts |
| El Salvador | Bolivia | Santa Lucia | Suriname |
| Brazil | Ecuador | Mexico | Turks and Caicos |
| Trinidad | Uruguay | Venezuela | Saint Vincent |
| Guiana | Dominican Republic | Virgin Islands | |

Survey data.

The choice of period aims to study the most recent context of income convergence in Latin America. Furthermore, the sub-periods analyzed in the later sections refer to changes in the LA economic cycles, with 1995–2000 being a period of adjustments and low growth, as well as the implementation, in some countries in the region, of neoliberal approaches; 2000 to 2013 represents a period of greater growth driven by exports of primary products; and 2013 to 2017 represents a more recent period in which economic and political instabilities spread among countries in the region.

The choice of using data up until 2017 stems from a couple of significant reasons. Firstly, complete and reliable data for years beyond 2017 is not uniformly available for all countries. This limitation in data availability restricts the ability to conduct a comprehensive analysis across various countries and years. As a result, utilizing data up to 2017 ensures a more consistent and accurate representation of the global landscape.

Secondly, another pivotal factor influencing this decision is the sensitivity of estimations derived from the available data. The accuracy and reliability of estimates are notably contingent on the number of countries included in the dataset. When dealing with more recent data, the risk of excluding a considerable number of countries due to incomplete data increases. This potential loss of data can significantly impact the quality and reliability of any estimations or conclusions drawn from the analysis.

The data on the participation of each category of foreign trade in the product per capita are also from the UNCTAD database. The classification of exports and imports by category of technological intensity, on the other hand, follows the scale proposed by Lall (2000), which is a reference for studies dealing with typologies related to the technological components of exports and imports. The decomposition is compatible with a panel data technique, in which the calculation of the components $R(v)$ and $P(v)$ requires two observations on income for a data set.

### 3. Results

Regarding the pattern of exports, according to their technological content, there is a concentration of low-technology products and lack of diversity in Latin American foreign trade. Exports to the European Union, Asia–Oceania and, especially, China overwhelmingly consist of basic and primary agricultural or mineral products. For example, in the Chinese case alone, such products correspond to 93% of exports. This pattern of exports has historically characterized Latin America's foreign trade. Thus, despite the diversified context of Latin American countries in geopolitical, economic, and socio-cultural terms, Latin America has been unable to break with its characteristic agro-export model.

Thus, as Braga (2002) explains, regionally, the development process has been based on the exploration of natural resources and a limited productive cooperation: "This limit is explained by the colonization of the territory...based on a process of exploitation of natural

resources and slave labor (and later on cheap labor) controlled by regional oligarchies according to international hegemonic interests".

However, if, on the one hand, Latin American exports to the rest of the world are typically concentrated and based on raw materials, intraregional trade is more diversified, with a greater share of more complex, technologically intensive manufactures. Exports of medium-tech goods correspond, for example, to 33% of intraregional trade, followed by low-tech exports. High-tech exports also represent a considerable value.

An important case in the context of Latin American international trade is that represented by the United States, which is an important trading partner for Latin American exports that include a much wider range of goods in relation to other destinations. About 60% of exports to the USA are medium- or high-technology goods.

This is largely explained by the context of Mexican exports to the United States in the spatial context of the North American Free Trade Agreement (NAFTA)[1]; for a better description of this relationship, Lima and Lo Turco (2010) is recommended.

Thus, as noted, exports to markets outside Latin America, in general, are quite concentrated and low-tech. On the other hand, the pattern of exports to intraregional markets is more diversified, with a higher share of manufactured goods. Figures 1 and 2 show this pattern is also repeated in some cases where there are integration schemes.

In the case of the countries within the Southern Common Market and the Andean Community, the pattern of concentration of exports to the rest of the world is much more evident, with basic products, whether agricultural or mineral, representing around 50% of exports. The countries of the Central American Common Market and the Caribbean Community have a more diverse pattern of exports compared to the rest of the world, although the total volume of exports is smaller than that of Latin America. Even though the quantity of agricultural and primary goods is strong in those regions, it is less than in the South American countries.

By contrast, when analyzing intraregional exports, integration schemes have a more diversified range of exports with more manufactured goods with a greater technological component, especially in the case of the Mercosur and MMCA countries.

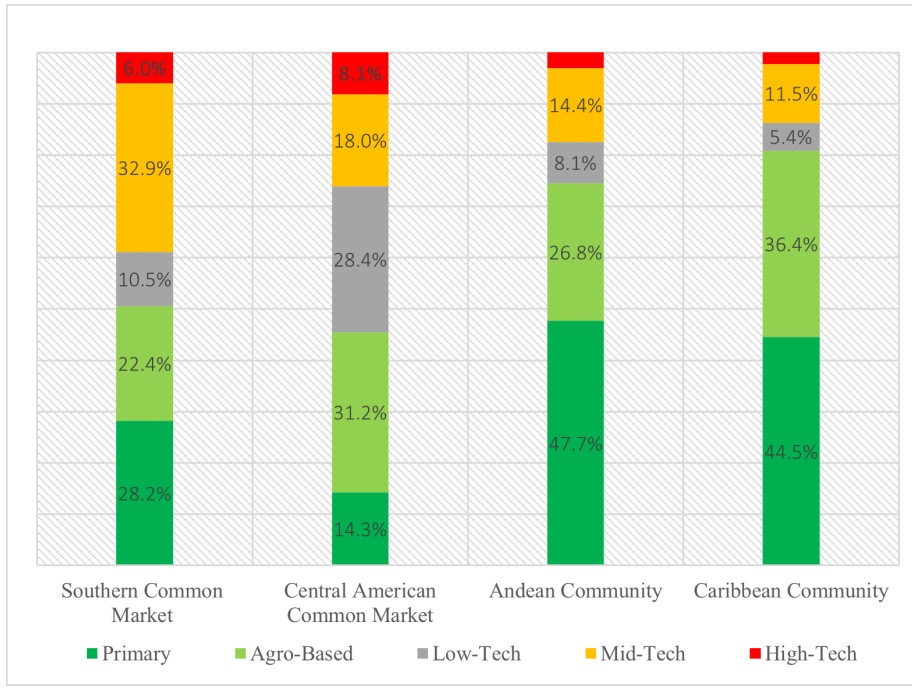

**Figure 1.** Intraregional exports by selected blocks.

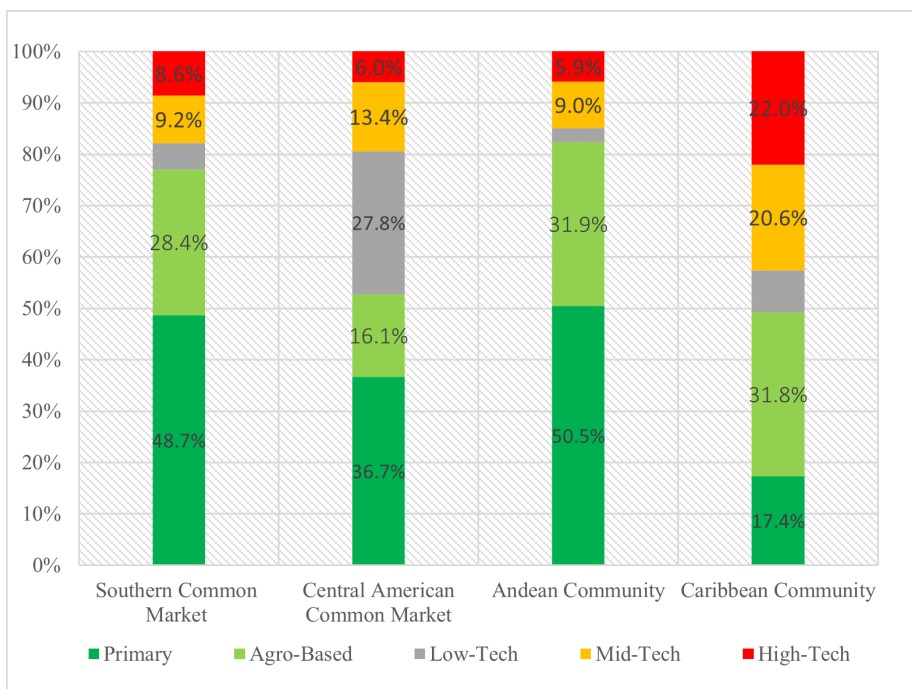

**Figure 2.** Extraregional exports by selected blocks.

Many implications can be drawn from these data; the first is that Latin America, despite the attempts at industrialization undertaken in previous decades, has not effectively managed to substantially change its role in the international division of production, remaining essentially as an exporter of food and raw materials, a set-up it has historically held in international trade. Thus, as Cortada (2007) points out, though much of Latin America has changed its productive structure and developed, "its continues to be inserted peripherally, dependent on the patterns of accumulation generated in the central economies".

On this subject, more information can be added to the debate, for example, regarding specific countries, such as Mexico, where the export pattern is centered on manufactured products, which is quite unique in the general Latin American context. Also, the changes induced by industrialization within the productive structure of the countries has allowed new patterns of employment and domestic consumption.

Finally, it is also possible to add the creation of sources of export dynamism, such as the potential of intraregional markets, as a way to diversify and industrialize the exports of the countries in the region, even if the full potential has not yet been effectively exploited, as shown by Lima and Lo Turco (2010) in their exercises to calculate the potential of this trade for the countries in the region and how much of that potential is actually used.

Regarding that potential, it is noted that intraregional trade, strengthened by integration, can, in fact, contribute to a more dynamic pattern of exports, since they tend to be centered on manufactured goods with greater technological content that tend to boost local economies. However, as Cortada (2007) observes, the subordination and historical dependence of Latin America in relation to the central countries "restricts the possibility of intraregional integration in every way". This explains the reduced exploitation of the potential of intraregional exports, as shown in Lima and Lo Turco (2010), since the ties to the central economies are closer than those among the countries of the region.

In the set of graphs, one can see the trajectory of the development of inequality in product per capita among the 39 Latin American countries and territories and some of their subspaces selected for this paper. Over the period, there was an 18.5% reduction in inequality in Latin America.

It is important to note that this reduction in the distribution of product per capita reflects the context of convergence or divergence among the sub-regions. Importantly, this reinforces the fact that these subspaces do not necessarily undergo the same process

of asymmetry reduction. A case in point is the Mesoamerican region, which saw a 20% increase in asymmetry. This reinforces the evidence found in the literature, in studies such as Barrios et al. (2019), who affirm that Latin America is marked by convergence clubs with different steady-state patterns and different convergence speeds, or even divergence in some of these clubs. Thus, the reduction in total dispersion only reflects the average interactions of these patterns.

In the other selected subspaces, there was a reduction in inequality. Regarding the magnitude of the dispersion, the highest level in inequality is found in the Caribbean region, where countries and territories whose economies, based on banking and tourism services, such as the Bahamas, Cayman Islands, and British Virgin Islands, present a high product per capita by Latin American standards. However, in that same region, there are countries with the lowest standards of development in Latin America, such as Haiti, even though this difference shows a declining pattern.

Finally, the South American countries have the lowest intraregional inequality, with a coefficient of 0.28. The reduction in inequality between South American countries was also around 20%. But these data reinforce some results found in the literature that have shown that South American countries, especially Argentina, Brazil, Chile, and Uruguay, have a convergence of their product per capita, as demonstrated by Barrios et al. (2019).

Additionally, the convergence has occurred faster than in the other clubs, according to Barrios et al. (2019). This defines South America as one of the subspaces of Latin America with the lowest rates of dispersion among its countries, as shown in the graphs above. In addition, this convergence is closely associated with the integration schemes present in South America, as observed by Müller (2008).

It is interesting to note how inequality reflects the region's economic fluctuations. In the context of the Argentine economic collapse in 2001–2002, as well as the Brazilian crisis, there was a sharp retraction in the inequality coefficient, due to the abrupt reduction in the product per capita of the richest economies in the region until then, leading to strong reduction in total inequality.

## 4. Discussion

The data below (Table 3) are composed of the results of the decomposition of the variation in $\sigma$-convergence, expressed by the inequality coefficient, as shown in the structure in Section 2.1. This decomposition is conducted using the various components of the convergence process. The lines refer to the selected sub-periods; the second column shows the change in the inequality coefficient and measures the $\sigma$-convergence; the fourth and sixth columns present the respective contributions of the progressive growth of the product per capita ($\beta$-convergence) and of the reclassification or mobility (*RC*) towards the change in general inequality.

Most of the reduction in the variation in the inequality coefficient occurred between 2000 and 2017. During this period, Latin America entered its longest point of convergence within the selected time series for this paper. The reduction in the inequality coefficient ($\Delta$G) reached 19.4%, which resulted in a $\sigma$-convergence of −0.102. Within this variation, the mobility effect played a less important role; the reduction in inequality was mainly induced by $\sigma$-convergence, that is, by progressive growth. As a percentage of the variation in the inequality coefficient ($\Delta$G), the mobility effect represented only 1.6%. Thus, most of the growth process in Latin America, in this period, was marked by its progressive effect, which represented 21.0% of the $\sigma$-convergence. This means that the growth process has proportionally benefited the poorest economies in the region.

In the period, from 1995 to 2017, the variation in the inequality coefficient ($\Delta$G) was −16.5% (a $\sigma$-convergence of −0.083), that is, the reduced dispersion observed between 2000 and 2017 was balanced by a period of increased dispersion at the beginning of the series, between 1995 and 2000. Note that, in this period, the divergence was 0.019. However, for the entire period under study, there is still evidence of both $\sigma$-convergence and $\beta$-convergence. For the most recent period (2013 to 2017), there is a slight convergence, though it is very

small and its $\rho$-value is statistically insignificant, making further conclusions impossible. Thus, although there is still convergence, it is much less intense than that seen in earlier periods.

**Table 3.** Income Convergence.

| Period | $\sigma$-Convergence ($\Delta$G) | $\rho$-Valor | $\beta$-Convergence- ($\beta$C) | $\rho$-Valor | Mobility (RC) | $\rho$-Valor |
|---|---|---|---|---|---|---|
| 1995–2017 | −0.083 | 0.000 | −0.098 | 0.000 | 0.015 | 0.027 |
| 1995–2000 | 0.019 | 0.134 | 0.015 | 0.223 | 0.003 | 0.057 |
| 2000–2017 | −0.102 | 0.000 | −0.110 | 0.000 | 0.008 | 0.039 |
| 2013–2017 | −0.006 | 0.401 | −0.010 | 0.188 | 0.004 | 0.159 |

Survey data.

Again, regarding the data above, it is interesting to note the level of significance in each period. In all periods, $\sigma$-convergence is significant at a level of at least 5%, which is valid for the period from 2000 to 2017, when most of the convergence between Latin American countries occurs. Between 1995 and 2000, when there is a greater dispersion of the product per capita, the $\rho$-value is not significant. In the period of 2013 and 2017, marked by the stagnation in important economies in Latin America, there is still a trend towards convergence, although $\rho$-value is not statistically significant.

The literature demonstrates that, after the mid-1990s, economic growth is unequivocally characterized by unconditional convergence, a phenomenon valid even for Latin America. Consequently, studies with significantly longer analysis periods, particularly those commencing in the 1960s or 1970s, generally reveal no evidence of a reduction in dispersion ($\sigma$-convergence) among Latin America's per capita income Delbianco and Dabús (2019); Dobson and Ramlogan (2020); King and Ramlogan-Dobson (2016). Nevertheless, more recent investigations focused on income convergence post-1990s already indicate substantial evidence of convergence within the region Cunha et al. (2011); Patel et al. (2021); Paulo (2020).

As a general conclusion, it is noted that both methods are consistent. In addition, there is a reduction in the dispersion of the product per capita in the analyzed period. The objective now is to better understand what role foreign trade has played in reducing this dispersion, or, in other words, as has now been established, in the $\sigma$-convergence. The main justification for using the alternative approach is exactly its advantage in allowing the variation in $\sigma$-convergence to be decomposed according to sources. In this specific case, these sources will be the components of aggregate demand, selected from special categories of foreign trade. This is because the sources linked to foreign trade weigh considerably in variation of regional inequality, in addition to being a source of an ever more volatile nature and conditioning important elements when plotting regional economic policy.

*Decomposing Convergence Arising from Foreign Trade*

This section describes the main results obtained from the proposed method, which show the effects foreign trade has had on the $\sigma$-convergence of Latin American countries and territories. Thus, $\sigma$-convergence is decomposed from the components of aggregate demand, with trade balances expanded according to selected categories. The results are shown in the Tables A1 and A2. The first column lists the types of components of aggregate demand, while the second and third columns show the composition effect and concentration, respectively. Finally, in the last column, we find the total effect of the contribution of each component in the variation of inequality.

As shown above, the total variation of the coefficient was $\Delta$G = −16.7%, which can also be interpreted as $\sigma$-convergence, that is, the variation in the dispersion between the set of countries. Again, it should be noted that the components of internal absorption

(household and government consumption, in addition to gross capital formation) contribute considerably to the regional convergence of product per capita. An initial synthesis of the data shows that exports and imports have important effects on the evolution of convergence, with exports corresponding to 40.6 and imports to $-55.2$ percent of the total $\sigma$-convergence. Thus, the various trade policies and integration strategies, by affecting the volume and patterns export and import specialization, also significantly affect convergence.

Nevertheless, although exports and imports significantly affect convergence, their effects do not occur in the same manner. While exports tend to reduce dispersion between countries, that is, to increase convergence, imports are one way in which dispersion tends to increase. In this paper's time frame, of the reduction of 16.7% found in the dispersion of the product per capita, about 40% occurred due to variations in the export accounts. On the other hand, imports, with a participation of around 50%, significantly contributed to increased dispersion.

The importance of this paper is based, precisely, on opening these accounts to better explain how these effects are associated with the values mentioned above. Thus, a cross-section of foreign trade is taken according to patterns of export and import specialization in terms of technological content and categories of intraregional and extra-regional trade. The data are analyzed below.

Both the exports of goods and services contributed towards increased regional convergence, despite the fact that the exports of goods contributed towards increased regional dispersion, that is, to decreased convergence, by the composition effect. However, this is more than offset by the participation effect. By contrast, imports, whether of goods or services, contributed to the dispersion of the product per capita, due to both effects, given the regressive character of this source on convergence.

In the case of the variations associated with the technological content of Latin American international trade, one can see the progressive measures associated with exports, regardless of their technological structure, are mostly progressive, that is, they contribute to increasing convergence. An exception is the exports of services. This is due to the fact that this export segment is highly represented in the product, at around 35%, although, in a very heterogeneous way between countries, the source concentration index corresponds to 0.7586, indicating that few Latin American countries are specialized in service exports. Moreover, this category contributed to the convergence, precisely because both its participation and its concentration coefficient decreased over the period, which favors general convergence.

Again, regarding the results found for service exports, one can see that their regressive effect is dominant in the total accounts. Thus, total exports are regressive. As stated before, service exports weigh heavily in the product per capita and, consequently, in the total of goods and services exported. Thus, their regressive effect becomes dominant. Hence, this demonstrates the importance of opening trade accounts, as it is of fundamental interest to know what is in fact regressive or progressive in each subcomponent and which are dominant.

Service exports require a more complex and competitive production structure, which is still poorly developed in Latin American countries. This explains the high degree of concentration of this source, with only a few countries in the region meeting the conditions to specialize in exports of this type. However, exports of goods, whatever the subcategory, are progressive, but do not have a dominant effect on this balance. Based on these results, it would be interesting, regarding policy recommendations, if there were an agenda that allowed the poorest countries to become more competitive in the production and export of services, while, at the same time, in which all of Latin America could seek to expand its markets for goods exports.

On the other hand, regarding imports, one can see that the main categories, those with the largest share in the product per capita, are service imports with 18%, medium-technology goods at 10%, and those based on agricultural products with around 8% of the product per capita. This category, except for agricultural products, has a well-defined

regressive pattern and contributes to increasing the dispersion of regional products among countries. Importantly, the import categories with greater weight in the Latin American product are shown to be regressive in terms of convergence. The other import categories are progressive, but have a smaller share in the total product and, therefore, have less effect on reducing their dispersion.

From the point of view of an ideal import policy to promote convergence and integration, the agenda would need to seek to reduce the deficit in transactions of medium-tech goods, since Latin America imports many more goods in this category than it manages to export, while it is also the basis of regional imports. This should come with a plan to reduce the volume of service imports. Another way to achieve a similar result would be to promote greater access by the poorest countries to international markets for services and medium-tech goods, increasing total imports and putting pressure on the trade balance.

The Table A2 describes the $\sigma$-convergence decomposition for components of aggregate demand, but the foreign trade accounts are displayed in terms of intraregional trade. Analysis of the progressiveness measures supports the perception that intraregional trade favors income convergence in Latin America. Both exports and intraregional imports are progressive. In the case of extra-regional trade, as expected, exports are progressive, since, as shown in the previous table, exports are strongly progressive towards convergence. On the other hand, extra-regional imports are regressive.

However, despite the progressive nature of intraregional exports, there has been a reduction in their participation in the Latin American product per capita. This goes against efforts designed to promote trade integration. Thus, the integration schemes seen in the region are not as effective at consolidating intraregional trade, at least in terms of its growth as a proportion of the product. For this reason, Latin American intraregional exports showed a composition effect of $-4.3\%$, pressuring for greater dispersion (since this source is progressive), but reducing its share in the product.

The composition effect of intraregional exports is offset by their concentration effect, that is, although this type of progressive trade has required participation in the product, it has become even more uniform among the countries. The concentration coefficient of this source decreased by 12.2%, which led to a concentration effect of 13.5% in the decomposed convergence. As a result, the net effect was 9.2%, still favoring convergence, but much less than its potential. Thus, to some extent, the potential for the convergence of the regional economies is wasted by the absence of a policy that effectively promotes the dynamism of intraregional exports, which might lead to the growth of this source as a proportion of the product.

These data complement and support those observed by Lima and Lo Turco (2010), who also identified underutilization of intraregional trade in Latin America in terms of potential gains actually obtained. Thus, there is the added fact of the underutilization of this type of trade, not only in its scales, but also in its effect on convergence and, consequently, on Latin American integration.

In the case of extra-regional exports, a similar process occurs with the loss of their participation as a proportion of the product, while they are also more uniform among countries. This reinforces the perception that Latin America experienced a period of loss of dynamism in its exports in 1995 and 2017, contributing to a negative composition effect, only offset by the reduced concentration coefficient in both sources.

Finally, it should be noted that the concentration effect dominates the variation of total convergence, both in the case of convergence composed of technological intensity and of trade. However, the composition effect shows a negative variation, that is, it contributes towards increased dispersion by $-9.47\%$; the concentration effect more than dominates the total variation of convergence. This is, obviously, as expected; one does not see a profound change in the composition of the components of aggregate demand in the total product per capita, whatever the parameter for decomposition, as mentioned above.

A final notable fact is the intensity of the measure of progressiveness of intraregional and interregional imports. In the case of the former, the intensity is much lower, which

indicates a relationship of strong progressivity in this category. Thus, this pattern of trade based on the meeting of domestic consumption demands in Latin America by other countries in the region tends to greatly increase regional convergence. In the case of the latter, the concentration ratio of interregional imports is quite high, which indicates that the source has a strong tendency towards regressivity. Thus, when the domestic needs of Latin America are met by countries outside the region, this tends to increase product dispersion.

## 5. Conclusions

This paper has analyzed the hypothesis of convergence of the product per capita between Latin American countries in the period 1995–2017. Notably, to do so, it adopts an alternative method of identifying convergence. A characteristic of this approach is that it allows the well-known $\sigma$-convergence measure to be decomposed in two ways. One makes it possible to determine the participation of the growth progressivity component (or $\beta$-convergence) and the participation of the degree of growth mobility. The other facilitates the decomposition of $\sigma$-convergence into product strengths, which, in this paper, are components of aggregate demand, as they reflect different aspects of national economic policies.

The first observation that can be made is that absolute convergence occurred for the sample of Latin American countries and territories selected in the 1995–2017 period. The convergence was on the order of 16.7% and became more intense after 2004, when Latin America entered a phase of greater economic growth; however, it decelerated in the most recent period, which was characterized by a phase of lower growth. Thus, Latin American convergence tends to accelerate in cycles of greater growth. Additionally, this result was confirmed, both by using the proposed alternative approach as well as by the traditional measures of convergence, while also corroborating recent findings reported in the literature.

Regarding the forms of decomposition, it was noted that the recent growth in Latin America was, above all, of the progressive type, that is, it was characterized by a higher growth rate of economies with lower product per capita. These data are in line with what is found in the descriptive data, noting that these countries had a higher growth rate. Thus, the modality effect of growth, although present, has no relevant effect.

In the decomposition by sources of growth, the components of foreign trade are found to be relevant in determining $\sigma$-convergence, with exports contributing to the convergence of the product per capita, whereas imports contribute to its divergence. When the trade accounts are considered according to technological content, there is no significant influence of a specific category; all are progressive and help to reduce regional inequality, regardless of the amount of technology incorporated. However, exports of services have a regressive typology, due to the specializations required in the productive structure for a country to achieve a relevant result in terms of its exports of services, a phenomenon that still seems distant for most Latin American countries.

In the case of imports, the main categories in terms of share in the product, whether goods or services, are regressive. Thus, imports typically contribute to increase regional patterns of divergence. However, when considering the opening of accounts in relation to the destination of imports, it is noted that those transacted in the intraregional market contribute to increase convergence, as do intraregional exports.

Regarding the paper's limitations, the approach used in this paper is non-parametric, which limits the possibility of testing the robustness of the results to check their statistical consistency. In addition, a part of the time dynamics is lost when only the variatiosn in an initial and an end time are analyzed. Considering the recommendations for further research, more in-depth theoretical and analytical studies are needed on how these selected categories actually affect convergence. Here, we have focused on analyzing the magnitude of these effects, rather than the ways in which they were processed. New ways of extending the data by selecting other decomposition components are also very welcome.

We left some policy recommendations for intensification of economic integration strategies in Latin America. Intraregional trade can indeed contribute to promoting income

convergence between countries in the region. Many recommendations for further studies arise from the data obtained here. The contribution of different business sectors in relation to inequality would be one of the most outstanding. Observing how the different sectors drive/boost/dynamize convergence provides valuable information on this topic. The analysis of longer periods is also highly recommended.

**Author Contributions:** Conceptualization: E.M.P. and O.T.S.; Methodology: E.M.P. and O.T.S.; Validation: E.M.P. and O.T.S.; Formal analysis: E.M.P. and O.T.S.; Resources: E.M.P. and O.T.S.; Writing—original draft: E.M.P. and O.T.S.; Writing—review and editing: E.M.P. and O.T.S. All authors have read and agreed to the published version of the manuscript.

**Funding:** This research was funded by Higher Education Personnel Improvement Coordination (CAPES, in Portuguese) and Pontifical Catholic University of Rio Grande do Sul.

**Informed Consent Statement:** Not applicable.

**Data Availability Statement:** Data are available upon request.

**Acknowledgments:** Higher Education Personnel Improvement Coordination (CAPES, in Portuguese) and Pontifical Catholic University of Rio Grande do Sul.

**Conflicts of Interest:** The authors declare no conflicts of interest.

## Appendix A

**Table A1.** Decomposition of convergence by technological intensity.

| Aggregate Demand | Composition Effect | Concentration Effect | Total Effect | Type |
|---|---|---|---|---|
| Household Consumption | 0.55 | 71.70 | 72.25 | Progressive |
| Government Purchases | 0.04 | 19.22 | 19.26 | Progressive |
| Capital Formation | 0.12 | 22.95 | 23.07 | Progressive |
| **Exports** | −8.20 | 48.82 | 40.63 | Regressive |
| Exports of goods | −10.03 | 33.80 | 23.78 | Progressive |
| Primary exports | −6.68 | 5.95 | −0.73 | Progressive |
| Agricultural exports | −6.21 | 24.47 | 18.27 | Progressive |
| Low-tech | −1.92 | −0.06 | −1.98 | Progressive |
| Med-tech | −0.27 | 0.04 | −0.23 | Progressive |
| High-tech | 0.37 | 0.34 | 0.71 | Progressive |
| Unclassifiable | 4.68 | 3.06 | 7.74 | Progressive |
| Service exports | 1.83 | 15.02 | 16.85 | Regressive |
| **Imports** | −2.51 | −52.69 | −55.20 | Regressive |
| Imports of goods | −1.12 | −37.56 | −38.68 | Regressive |
| Primary imports | −0.27 | −9.58 | −9.85 | Progressive |
| Agricultural imports | 0.04 | −10.40 | −10.36 | Progressive |
| Low-tech | 0.02 | −5.12 | −5.10 | Progressive |
| Med-tech | −0.63 | −3.35 | −3.98 | Regressive |
| High-tech | 0.24 | −7.94 | −7.70 | Progressive |
| Unclassifiable | −0.52 | −1.17 | −1.69 | Regressive |
| Service Imports | −1.39 | −15.13 | −16.52 | Regressive |
| Total | −10.0 | 110.0 | 100 | - |

Survey data.

**Table A2.** Decomposition of convergence by type of trade.

| Aggregate Demand | Composition Effect | Concentration Effect | Total Effect | Type |
|---|---|---|---|---|
| Household Consumption | 0.55 | 71.70 | 72.25 | Progressive |
| Government Purchases | 0.04 | 19.22 | 19.26 | Progressive |
| Capital Formation | 0.12 | 22.95 | 23.07 | Progressive |
| **Exports** | −13.95 | 54.58 | 40.63 | Regressive |
| Exports of goods | −15.78 | 39.56 | 23.78 | Progressive |
| interregional Exports | −11.52 | 26.11 | 14.59 | Progressive |
| intraregional exports. | −4.26 | 13.45 | 9.19 | Progressive |
| Exports of goods | 1.83 | 15.02 | 16.85 | Regressive |
| **Imports** | 3.77 | −58.97 | −55.2 | Regressive |
| Imports of goods | 5.16 | −43.84 | −38.68 | Regressive |
| interregional Imports | 0.87 | −7.59 | −6.72 | Regressive |
| intraregional Exports | 4.29 | −36.25 | −31.96 | Progressive |
| Service Exports | −1.39 | −15.13 | −16.52 | Regressive |
| Total | −9.47 | 109.48 | 100.01 | − |

Survey data.

**Table A3.** Selected foreign trade categories.

| Foreign Trade Categories |
|---|
| Internal absorption = Consumption + Investments + Government Purchases |
| Total Exports |
| Total exports of goods |
| Primary basic intraregional goods |
| Agricultural basic intraregional goods |
| intraregional basic goods ores |
| Intraregional low-technology manufactured goods |
| Intraregional medium-technology manufactured goods |
| intraregional high-tech manufactured goods |
| Interregional primary basic goods |
| Agricultural interregional basic goods |
| Interregional basic goods ores |
| Interregional low-technology manufactured goods |
| Interregional medium-technology manufactured goods |
| Interregional high-tech manufactured goods |
| unclassified goods |
| Total service exports |

**Table A3.** *Cont.*

| Foreign Trade Categories |
| :---: |
| Total Imports |
| Total imports of goods |
| Primary basic intraregional goods |
| Agricultural basic intraregional goods |
| Intraregional basic goods ores |
| Intraregional low-technology manufactured goods |
| Intraregional medium-technology manufactured goods |
| intraregional high-tech manufactured goods |
| Interregional primary basic goods |
| Agricultural interregional basic goods |
| Interregional basic goods ores |
| Interregional low-technology manufactured goods |
| Interregional medium-technology manufactured goods |
| Interregional high-tech manufactured goods |
| unclassified goods |
| Total imports of services |

Survey data.

## Note

[1] The decision to support the replacement of NAFTA took place in November 2018 and was signed by representatives of the three countries. Because it represents an update of the old agreement, the USMCA is also called NAFTA 2.0. However, the approval of the legislation of each country is still necessary to enter into force.

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
