# Peer review of "Foreign Trade and Income Convergence in Latin America"

_economies, doi:10.3390/economies11090235_

Round 1

Reviewer 1 Report

1. Key concepts used for analysis should be elaborated in more detail and discussed why these concepts are relevant for the analysis in this paper, notably:

l  Composition effect and Concentration effect; and

l  Low-, middle- and high-tech export/import     

 2. Figure 1 should add a note of the abbreviations.

Author Response

All suggestions have been incorporated.

1 Key concepts used for analysis should be elaborated in more detail and discussed why these concepts are relevant for the analysis in this paper, notably:  Composition effect and Concentration effect; and Low-, middle- and high-tech export/import

R: A new subsection (lines 159 to 188) was added to explain the concepts pointed out by the reviewer. It was also added from Table 1 to explain and exemplify the concept of technological intensity. Line 218.

2. Figure 1 should add a note of the abbreviations.

R: A note of abbreviations has been included. Line 258.

Reviewer 2 Report

> The paper explores the variation in inequality as a measure of > convergence or divergence in per capita income. The topic is interesting. > But there are some points need to be addressed before publication. > 1.The abstract is too short, you should make it longer, focusing more > finding for your paper. > 2.Why the data only using 1995-2017? if possible, using latest like 2022. > 3.For the 3rd part of results analysis, you should give more clear new > ideas by structuring it. > 4. Figure 1 is not clear enough, you can use colorful show. > 5. In the last part, you should give more policy implications and > suggestions. > 6. The references are too less, while some more new papers should be > added into the ref. eg: > Liu et al., 2023, Assessing the role of economic globalization on energy > efficiency: evidence from a global perspective. China Economic Review,77 > 101897.

Extensive editing of English language required。

Author Response

1.The abstract is too short, you should make it longer, focusing more > finding for your paper. >

R: Correction made

2.Why the data only using 1995-2017? if possible, using latest like 2022. >

R: The article researches convergence between countries. Therefore, a very broad database is required. Even though the UNCTAD data are quite up to date, this is not the case for two countries. Many countries in the sample do not have complete data. Updating the data for 2023 would imply excluding several countries, whose data are not complete. This would significantly compromise the results. In addition, the recent context of political, economic and health crisis requires further research than would be possible within the scope of this article.

3.For the 3rd part of results analysis, you should give more clear new > ideas by structuring it. >

R: Suggestion incorporated in part. Some points have been added to clarify some concepts. 

4. Figure 1 is not clear enough, you can use colorful show. >

R: Correction made.

5. In the last part, you should give more policy implications and > suggestions. >

R: Correction made.

6. The references are too less, while some more new papers should be > added into the ref. eg: > Liu et al., 2023, Assessing the role of economic globalization on energy > efficiency: evidence from a global perspective. China Economic Review,77 >

R: Suggestion incorporated in part. Some implications in terms of policy recommendations and other derived studies are pointed out. New references have also been added.

Reviewer 3 Report

In my view, the paper has the following limitations:

1. The paper is very ambitious and its scope is too broad. The paper deals with several very general and complex issues, and it is not clear to see what is the main research question underlying the research. Is it beta convergence? Is it sigma convergence? Is it the link foreign trade-convergence?  It would be much better to narrow done and define a much more specific research question and try to answer it in more detail. 

2. Linked to the above issue, I miss some paragraphs in the paper explaining to the reader what the recent literature has said about the object of research. The authors need to identify and clarify 1.  what we know already about the main topic they are covering, according to the research done by other authors), 2. in which aspects there is a gap in the literature (because there is something that the literature has not covered yet, o because there is controversy about an issue), and 3. what they are looking at specifically in order to fill this gap.  My impression is that beta and sigma convergence have been very much explored already in past decades, and hence it is difficult to find something new in this regard. If the paper really comes up with a very new result, it has to argue it very convincingly.

3. Partly because  the gap in the literature is not clearly identified, it is not clear to the reader, either, what the contribution of the paper is. In other words, what are the novel ideas and results this paper is offering? What is the paper showing which nobody before has shown?

3. Beta convergence: as defined by Barro and Sala i Martin (2004) and Sala i Martin (1996) and explored by the very abundant empirical literature on convergence, there is beta convergence when the point estimate of the variable capturing initial income or initial capital in a regression relating the growth rate with a set of regressors is negative and significant. Depending on how these regressors are defined, beta convergence may be absolute or conditional. To argue about the presence or absence of beta convergence in a sample the researcher must detail how this regression is specified. This information is not given in the paper. Table 3 details the results of "Barro regressions", but we do not know how these Barro regressions are specified. Do they use cross section data or panel data? Which regressors are included? How many observations are there in the estimation? How is heteroskedasticity (and probably, endogeneity as well) handled? This information is key in order to conclude if there is or there is not beta convergence in a sample. In addition, convergence is a concept developed in the framework of growth theory, which by definition deals with the long run (not with the short run business cycle). The periods 1995-2000 and 2013-2017 are too short to let us get any conclusion about growth in the long run (as opposed to performance over the cycle)

4. Inequality and sigma convergence: It is not clear to me if the paper is focusing on inequality within each country in the sample or inequality between the different countries in the sample. Is it looking at the degree of inequality among the citizens of Argentina, for example, or at the degree of inequality between Argentina, Brazil and Peru considered as a group? This distinction is very important. Usually the literature has explored inequality within a country (among Argentinians, for example) by means of the Gini coefficient; sigma convergence, instead, is defined as the dispersion displayed by a variable, (such as growth, income, per capita income, and so forth), within a group of geographical units that are more or less comparable (countries, regions, provinces...). If there is a lot of dispersion in per capita income for a sample of countries, then we can conclude that there is not sigma convergence in that sample. It is not clear to me what the paper is looking at when exploring inequality. Since the paper is looking at a sample, it seems to be exploring sigma convergence, but if this is the case the contribution of Section 2.1. does not fit with the rest of the paper. Moreover, as discussed by Barro and Sala i Martin (2004) and Sala i Martin (1996), there is a very close connection between beta and sigma convergence in a sample (but not between Gini coefficients and beta convergence, since as discussed above Gini coefficients usually refer to one country while beta convergence refers to a group of countries or regions).

4 Section 4.1. is not clear enough, either. What is the theoretical framework which underlies the decomposition of convergence in composition effect and concentration effect?

5. Moreover, I do not think that the method used to obtain Table A.1. is very informative. The connection between foreign trade, technological intensity and convergence is a very broad and complex issue, which has to be analyzed by estimating a very detailed model, because the links between these aspects are in turn very convoluted (and these estimations, in turn, have to deal with a lot of econometric problems, as endogeneity, for example). Splitting sigma convergence in composition and concentration effect is just a descriptive technique which can be perhaps useful in teaching or in very general reports, but it does not seem rigorous enough to explore the links between foreign trade, technological intensity and convergence.

Barro, R., & Sala-i-Martin, X. (2004). Economic growth second edition. MIT Press

Sala-i-Martin, X. X. (1996). The classical approach to convergence analysis. The economic journal106(437), 1019-1036.

no comments

Author Response

1. The paper is very ambitious and its scope is too broad. The paper deals with several very general and complex issues, and it is not clear to see what is the main research question underlying the research. Is it beta convergence? Is it sigma convergence? Is it the link foreign trade-convergence?  It would be much better to narrow done and define a much more specific research question and try to answer it in more detail.

R: The main issue of the article is to explore the relationship between Latin American foreign trade and income convergence in the region. However, many other issues influence this relationship and need to be addressed minimally to have a more specific picture. By breaking down income convergence by trade categories, the main question of the article was answered.

2. Linked to the above issue, I miss some paragraphs in the paper explaining to the reader what the recent literature has said about the object of research. The authors need to identify and clarify 1.  what we know already about the main topic they are covering, according to the research done by other authors), 2. in which aspects there is a gap in the literature (because there is something that the literature has not covered yet, o because there is controversy about an issue), and 3. what they are looking at specifically in order to fill this gap.  My impression is that beta and sigma convergence have been very much explored already in past decades, and hence it is difficult to find something new in this regard. If the paper really comes up with a very new result, it has to argue it very convincingly.

R: The article's main contribution to the income convergence literature lies in its methodological approach. In fact, this topic has been extensively explored in the literature, but the methodological approach proposed in the article provides a strategy for decomposing income convergence, which is not so common in this type of study. Furthermore, this approach is quite flexible and allows income convergence to be approached from different perspectives. The study is actually an alternative to conventional income convergence methodologies, as proposed by NEILL and KERM (2008).

3. Partly because  the gap in the literature is not clearly identified, it is not clear to the reader, either, what the contribution of the paper is. In other words, what are the novel ideas and results this paper is offering? What is the paper showing which nobody before has shown?

R: The article intends to explore the relationship between foreign trade and income convergence in Latin America. For this, he proposes an alternative approach to measure and decompose income convergence. As mentioned before, convergence studies are numerous and there are several results, including for the Latin American case. The objective of the work is not exactly to show something new, but to explore more deeply the convergence of income through the perspective of foreign trade. In addition, its methodological approach allows several other studies, with other different strategies, to be developed.

4. Beta convergence: as defined by Barro and Sala i Martin (2004) and Sala i Martin (1996) and explored by the very abundant empirical literature on convergence, there is beta convergence when the point estimate of the variable capturing initial income or initial capital in a regression relating the growth rate with a set of regressors is negative and significant. Depending on how these regressors are defined, beta convergence may be absolute or conditional. To argue about the presence or absence of beta convergence in a sample the researcher must detail how this regression is specified. This information is not given in the paper. Table 3 details the results of "Barro regressions", but we do not know how these Barro regressions are specified. Do they use cross section data or panel data? Which regressors are included? How many observations are there in the estimation? How is heteroskedasticity (and probably, endogeneity as well) handled? This information is key in order to conclude if there is or there is not beta convergence in a sample. In addition, convergence is a concept developed in the framework of growth theory, which by definition deals with the long run (not with the short run business cycle). The periods 1995-2000 and 2013-2017 are too short to let us get any conclusion about growth in the long run (as opposed to performance over the cycle)

R: The purpose of Table 3 (Table 4 in the new version after revision) is to show that the results obtained with the proposed approach are similar to the well-known Barro and Sala i Martin regression. The intention is that this can give more support to the alternative approach. Thus, it would be unproductive to insert an entire section explaining Barro's regression estimation and its adjustment, given that this method is well known in the literature and that the data in Table 03 are not used to support the claims made in the article.

5. Inequality and sigma convergence: It is not clear to me if the paper is focusing on inequality within each country in the sample or inequality between the different countries in the sample. Is it looking at the degree of inequality among the citizens of Argentina, for example, or at the degree of inequality between Argentina, Brazil and Peru considered as a group? This distinction is very important. Usually the literature has explored inequality within a country (among Argentinians, for example) by means of the Gini coefficient; sigma convergence, instead, is defined as the dispersion displayed by a variable, (such as growth, income, per capita income, and so forth), within a group of geographical units that are more or less comparable (countries, regions, provinces...). If there is a lot of dispersion in per capita income for a sample of countries, then we can conclude that there is not sigma convergence in that sample. It is not clear to me what the paper is looking at when exploring inequality. Since the paper is looking at a sample, it seems to be exploring sigma convergence, but if this is the case the contribution of Section 2.1. does not fit with the rest of the paper. Moreover, as discussed by Barro and Sala i Martin (2004) and Sala i Martin (1996), there is a very close connection between beta and sigma convergence in a sample (but not between Gini coefficients and beta convergence, since as discussed above Gini coefficients usually refer to one country while beta convergence refers to a group of countries or regions).

R: According to the methodology proposed by NEILL and KERM (2008), the sigma-convergence has a very close connection with the Gini index, when calculated for a group of comparable geographic units. This allows performing broader decomposition calculations, such as the one proposed in this article. Section 2.1. has been reformulated to make this relationship more evident and fits in with the rest of the article.

6 Section 4.1. is not clear enough, either. What is the theoretical framework which underlies the decomposition of convergence in composition effect and concentration effect?

R: This issue was also pointed out by other reviewers. Thus, a new subsection was included in the article to clarify these concepts and how they are calculated.

7. Moreover, I do not think that the method used to obtain Table A.1. is very informative. The connection between foreign trade, technological intensity and convergence is a very broad and complex issue, which has to be analyzed by estimating a very detailed model, because the links between these aspects are in turn very convoluted (and these estimations, in turn, have to deal with a lot of econometric problems, as endogeneity, for example). Splitting sigma convergence in composition and concentration effect is just a descriptive technique which can be perhaps useful in teaching or in very general reports, but it does not seem rigorous enough to explore the links between foreign trade, technological intensity and convergence.

R: In fact, an important weakness of the proposed methodology is that it is not parametric. However, it is recommended to associate different approaches for greater accuracy of information. Here, as mentioned before, Barro's consolidated regression data are similar to the data found. An indication that the alternative approach is capable of producing consistent results.

Reviewer 4 Report

- data sets only until 2017...why??? data sources were used from UNCTAD data from 1995 to 2017, why did they end in this year? as it is 2023, this data may be outdated, in addition, extraordinary situations have occurred in the world in the past period, which significantly affected the researched area as well

- row 187-188 – its stated „....the study includes a sample of 38 Latin American nations and territories“ but later in row 264-265 is stated „...among the 39 Latin American countries and territories and some of their subspaces selected for this study.“ What is the correct number of countries selected for the study???

- row 220-221 – NAFTA??? What about USMCA??? I recommend supplementing the text with a paragraph explaining the changes in the NAFTA integration group, its demise in its original form and the creation of the USA-Canada-Mexico agreement, which replaced the original NAFTA agreement

- please clarify the selection of periods in table 2 more clearly

- writing style: in abstract and/or text you refer to this submission as essay (row 1, 68, 171, 516), later in introduction as article (row 44), and in conclusion as paper (row 480) and as well as study (row 516) - please use only one name, for example paper and correct the text

- sources used - the oldest source is from 1985 and the newest from 2019 - there are only 16 sources in total - from my point of view, this is insufficient for the current state of science, and it is necessary to mine the sources - I recommend at least 30 sources, while using the latest knowledge and using WoS or SCOPUS databases

- overall, the paper lacks a deeper theoretical basis, which would be appropriate to add

it is advisable to read the paper with a proofreader and unify the text

Author Response

- data sets only until 2017...why??? data sources were used from UNCTAD data from 1995 to 2017, why did they end in this year? as it is 2023, this data may be outdated, in addition, extraordinary situations have occurred in the world in the past period, which significantly affected the researched area as well.

R: The article researches convergence between countries. Therefore, a very broad database is required. Even though the UNCTAD data are quite up to date, this is not the case for countries. Many countries in the sample do not have complete data. Updating the data for 2023 would imply excluding several countries, whose data are complete. This would significantly compromise the results. In addition, the recent context of political, economic and health crisis requires further research that would be possible within the scope of this article.

- row 187-188 – its stated „....the study includes a sample of 38 Latin American nations and territories“ but later in row 264-265 is stated „...among the 39 Latin American countries and territories and some of their subspaces selected for this study.“ What is the correct number of countries selected for the study???

R: 39 countries. Correction made.

- row 220-221 – NAFTA??? What about USMCA??? I recommend supplementing the text with a paragraph explaining the changes in the NAFTA integration group, its demise in its original form and the creation of the USA-Canada-Mexico agreement, which replaced the original NAFTA agreement.

R: The article researches convergence between countries. Therefore, a very broad database is required. Even though the UNCTAD data are quite up to date, this is not the case for two countries. Many countries in the sample do not have complete data. Updating the data for 2023 would imply excluding several countries, whose data are not complete. This would significantly compromise the results. In addition, the recent context of political, economic and health crisis requires further research than would be possible within the scope of this article.

- please clarify the selection of periods in table 2 more clearly.

R: It was included a paragraph between lines 229 and 235 to explain this point.

- writing style: in abstract and/or text you refer to this submission as essay (row 1, 68, 171, 516), later in introduction as article (row 44), and in conclusion as paper (row 480) and as well as study (row 516) - please use only one name, for example paper and correct the text.

R: Correction made.

- sources used - the oldest source is from 1985 and the newest from 2019 - there are only 16 sources in total - from my point of view, this is insufficient for the current state of science, and it is necessary to mine the sources - I recommend at least 30 sources, while using the latest knowledge and using WoS or SCOPUS databases.

R: A new subsection has been added, providing more information and references. In addition, other parts of the text have been reworked and new fonts have been added.

Round 2

Reviewer 1 Report

Accept in present form.

Author Response

All suggestions were accepted.

Reviewer 3 Report

The revised version of the manuscript includes a brief new subsection, 2.1.1. However, this is insufficient, in my view, to make the paper publishable; the new version has not addressed other important weaknesses of the paper that were pointed out in my initial report.

In particular, the connection between the two parts of the paper, the analysis of sigma convergence and the role of foreign trade, continues to be unclear. Although I welcome the introduction of the new subsection, the new paragraphs are not enough to provide a clear and rigorous theoretical foundation for the exercise summarized in Tables A.1 and A.2. This theoretical foundation is especially important because it does not seem to be developed in the paper by O’Neill and Van Kerm (2008), in which the authors base their analysis. Moreover, the analysis of convergence is usually performed within growth theory, which is developed in an analytical framework which focuses in the long run and in the production side of the economy. The notion of aggregate demand, however, belongs to a macroeconomic model which describes the economy in the short run and from the point of view of demand (rather than supply) in the economy (i.e. households, government, rest of the world). I really cannot see what is the theoretical mechanism assumed by the paper that enables the transition of one to the other. The authors should specify more clearly what this theoretical mechanism is, detailing what the economic assumptions and the equilibrium conditions are.

Other points:

1.       I still do not see what is the goal of the paper and, therefore, its main contribution. The authors say in their response to my comments that the objective is “to explore more deeply the convergence of income through the perspective of foreign trade”. This is a very ambiguous claim, which has to be narrowed down.

2.       The paper lacks a discussion of the related literature on beta and sigma convergence for Latin-American countries. This discussion is crucial in order to show if the results in the paper are consistent with the findings of other contributions. The authors have to compare their results with those of other authors in order to stress their validity.

3.       In my view, the Barro regression should be clarified and detailed, more so because it is presented as a robustness test and as a confirmation of the results on sigma convergence. Obviously a long explanation of the general setting of the Barro regression is not necessary; what is needed, however, is that the paper specifies at least the following additional issues (together with the parameter estimates and their standard errors): which control variables are included in the equation, if the analysis is performed with cross country observations or panel data, what is the number of observations, what is the R2. Notice that the analysis described in the last part of the paper is non parametric; this implies that we do not know how robust the results are, how valid the numbers in table A.1 are, or if they are driven by outliers. In addition, the results are obtained comparing two particular years, 2017 and 1995 (rather than analyzing the whole period). These are important limitations of the paper which reduce the validity of its findings. The authors, therefore, should try overcome these flaws backing up their results as much as possible, by comparing them with the literature (see point 2 above) and by explaining very clearly all the details of the estimations performed (as in the case of the Barro regressions).

minor editing is necessary

Author Response

The revised version of the manuscript includes a brief new subsection, 2.1.1. However, this is insufficient, in my view, to make the paper publishable; the new version has not addressed other important weaknesses of the paper that were pointed out in my initial report.

In particular, the connection between the two parts of the paper, the analysis of sigma convergence and the role of foreign trade, continues to be unclear. Although I welcome the introduction of the new subsection, the new paragraphs are not enough to provide a clear and rigorous theoretical foundation for the exercise summarized in Tables A.1 and A.2. This theoretical foundation is especially important because it does not seem to be developed in the paper by O’Neill and Van Kerm (2008), in which the authors base their analysis. Moreover, the analysis of convergence is usually performed within growth theory, which is developed in an analytical framework which focuses in the long run and in the production side of the economy. The notion of aggregate demand, however, belongs to a macroeconomic model which describes the economy in the short run and from the point of view of demand (rather than supply) in the economy (i.e. households, government, rest of the world). I really cannot see what is the theoretical mechanism assumed by the paper that enables the transition of one to the other. The authors should specify more clearly what this theoretical mechanism is, detailing what the economic assumptions and the equilibrium conditions are.

R: A section was included where this connection is methodologically explored in greater detail. The central idea is that the alternative approach opens up the possibility of decomposing the variation of dispersion by sources (which justifies the choice of this approach over traditional ones). As foreign trade is a key variable in Latin American growth, and the region's trade patterns have been used as a basis to explain dependency theory, the decision is made to decompose this dispersion precisely by variables related to foreign trade. These variables are present in the basic equation of aggregate demand exactly in the format required by the methodological approach (which justifies the choice of the demand side to address the issue)

I still do not see what is the goal of the paper and, therefore, its main contribution. The authors say in their response to my comments that the objective is “to explore more deeply the convergence of income through the perspective of foreign trade”. This is a very ambiguous claim, which has to be narrowed down.

R: The main idea of the article is to first measure the income dispersion variation (in the article's approach, this is done through the method proposed by O'NEILL and KERM, which associates this variation with the so-called sigma-convergence). These results are presented in Table 3. Next, the proposal is to decompose this variation by selected categories of foreign trade (for reasons ranging from the role foreign trade plays in the region's economic growth to historical arguments explaining the region's development). These results are presented in Tables A1 and A2. Thus, the article aims to contribute to studies on convergence and presents as distinctive features to highlight its results from those already presented in the literature: i) the alternative approach on which the data is based; ii) the possibility to configure an income convergence decomposition scheme; iii) to preliminarily observe how foreign trade affects this convergence for the Latin American case. Some limitations arise from the proposed method, which deserve to be better calibrated. However, the article provides an insight, albeit preliminary, into how foreign trade has impacted the income dispersion variation in Latin America.

The paper lacks a discussion of the related literature on beta and sigma convergence for Latin-American countries. This discussion is crucial in order to show if the results in the paper are consistent with the findings of other contributions. The authors have to compare their results with those of other authors in order to stress their validity.

R: Income convergence studies for Latin America often tend to indicate no evidence of income convergence in the region. However, the literature shows that after the mid-1990s, economic growth is unequivocally characterized by unconditional convergence, even applying to Latin America. Hence, studies with considerably long analysis periods, especially starting from the 1960s or 1970s, generally demonstrate that there is no evidence of a reduction in dispersion (sigma-convergence) among Latin America's per capita income. Nonetheless, more recent studies examining income convergence after the 1990s already point to strong evidence of convergence in the region. In light of this, some comments pointing out these features of income convergence in the literature have been included throughout the text. The findings presented in this article closely resemble those found in the literature that analyze a more recent timeframe and utilize databases similar to those used in this research. This fact has been added to the text.

In my view, the Barro regression should be clarified and detailed, more so because it is presented as a robustness test and as a confirmation of the results on sigma convergence. Obviously a long explanation of the general setting of the Barro regression is not necessary; what is needed, however, is that the paper specifies at least the following additional issues (together with the parameter estimates and their standard errors): which control variables are included in the equation, if the analysis is performed with cross country observations or panel data, what is the number of observations, what is the R2. Notice that the analysis described in the last part of the paper is non parametric; this implies that we do not know how robust the results are, how valid the numbers in table A.1 are, or if they are driven by outliers. In addition, the results are obtained comparing two particular years, 2017 and 1995 (rather than analyzing the whole period). These are important limitations of the paper which reduce the validity of its findings. The authors, therefore, should try overcome these flaws backing up their results as much as possible, by comparing them with the literature (see point 2 above) and by explaining very clearly all the details of the estimations performed (as in the case of the Barro regressions).

R: An equation was estimated for each period, resulting in a total of four equations. In all cases, the number of observations remained the same, 39. The data is analyzed using ordinary least squares estimation based on the cross-section formed, where the dependent variable is (1/T)*log(yiT/yi0) and the explanatory variable is logyi0. For the sake of simplification, no controls were included. The data is presented below.

Coefficient

T

p-value

R2

-0.006014

-3.08

0.004

0.2036

0.005191

1.32

0.196

0.0448

-0.006857

-4.16

0.000

0.3182

-0.001536

-1.74

0.089

0.0760

The purpose of this table is solely to demonstrate a similarity between the results of the two approaches, which can be clearly observed. Readers and evaluators may request further exploration into the Barro regression estimation. However, this was not the intention of this article. The two approaches differ significantly in terms of estimations, and the contribution of this article is to present results using a non-traditional approach for the Latin American case. We acknowledge that we had a rather ambitious idea of presenting the Barro regression as a means of validating the data presented in the article. Additionally, we are aware that convergence results are highly sensitive to the estimation method, controls, and model treatments applied, as well as the database used, among other components (as extensively shown in the literature). For these reasons, we have chosen not to include Table 4 in the article, but rather to present it to the evaluators. This approach avoids unnecessary complications and misinterpretations of the data. The data presented in Table 3 (the alternative approach) have undergone parametric tests confirming their validation and have been presented in the literature with relative success by O’NEILL and KERM. The data presented in Tables A1 and A2 merely decompose the sigma-convergence in Table 3, which has been parametrically tested.

Reviewer 4 Report

the paper is better after revision, but I would add (as stated in the reply to my comments) an explanation of the time period - i.e. a justification for why it is only up to 2017 (somewhere in sections 222-235)

also, I still have a problem with the citation of all sources - the article uses data from UNCTAD but there is no mention of them in the references

Author Response

The paper is better after revision, but I would add (as stated in the reply to my comments) an explanation of the time period - i.e. a justification for why it is only up to 2017 (somewhere in sections 222-235).

R: The suggestions provided have been incorporated into the article. Additionally, further corrections have been implemented: i) additional references have been included, and the reference section has been better organized; ii) a new section has been added to better present the concepts used in the article.

Also, I still have a problem with the citation of all sources - the article uses data from UNCTAD but there is no mention of them in the references

R: The suggestions provided have been incorporated into the article.

Round 3

Reviewer 3 Report

In my view, the paper may be published.